# Formulation and Functional Characterization of a Cannabidiol-Loaded Nanoemulsion in Canine Mammary Carcinoma Cells

**DOI:** 10.3390/pharmaceutics17080970

**Published:** 2025-07-26

**Authors:** Francisca J. Medina, Guillermo Velasco, María G. Villamizar-Sarmiento, Cristian G. Torres, Felipe A. Oyarzun-Ampuero

**Affiliations:** 1PhD Program in Agriculture, Forestry, and Veterinary Sciences, Universidad de Chile, Santiago 8820808, Chile; francisca.medina@veterinaria.uchile.cl; 2Centralized Laboratory for Veterinary Research (LACIV), Faculty of Veterinary and Animal Sciences, Universidad de Chile, Santiago 8320000, Chile; 3Drug Delivery Laboratory, Department of Pharmaceutical Science and Technology, Faculty of Chemical and Pharmaceutical Sciences, Universidad de Chile, Santiago 8320000, Chile; magabrielavillamizarsarmiento@gmail.com; 4Department of Biochemistry and Molecular Biology, Faculty of Chemical Sciences, Universidad Complutense de Madrid and Instituto de Investigación Sanitaria San Carlos (IdISSC), 28040 Madrid, Spain; gvelasco@quim.ucm.es; 5Center of New Drugs for Hypertension and Heart Failure (CENDHY), División de Enfermedades Cardiovasculares, Pontificia Universidad Católica de Chile, Universidad de Chile, and Universidad Andrés Bello, Santiago 8320000, Chile; 6Department of Clinical Sciences, Faculty of Veterinary and Animal Sciences, Universidad de Chile, Santiago 8320000, Chile

**Keywords:** cancer, canine mammary carcinoma, cannabidiol, nanoemulsion, drug delivery

## Abstract

**Background/Objectives**: Mammary carcinoma is a common disease in female dogs. Cannabidiol (CBD) can inhibit cell proliferation and induce apoptosis in human cancer cells. However, its low solubility in aqueous media requires solvents such as ethanol or dimethylsulfoxide that limit their dosage. Incorporating CBD into oil-in-water nanoemulsions (Nem) can improve its aqueous dispersibility. This study aimed to develop a CBD-Nem formulation and evaluate its effects on canine mammary cancer cell lines (CF41.Mg and IPC366) and non-cancer cells (MDCK). **Methods**: CBD-Nem was prepared with Miglyol 812 oil and Epikuron 145 V as the surfactant, and was characterized by analyzing size, morphology, zeta potential, release profile, and uptake/internalization. Moreover, the antitumor effects of CBD-Nem were evaluated in cancer cells through viability, proliferation, cell cycle, and migration–invasion assays. **Results**: CBD-Nem exhibited a monodisperse nanometric population (~150 nm), spherical shape, and negative zeta potential (~−50 mV). The in vitro release kinetics showed slow and sustained delivery at both pH 5.5 and pH 7.4. Rhodamine-Nem, as a fluorescent model of CBD-Nem, was taken up and homogenously internalized in CF41.Mg cells. CBD-Nem decreased the viability of cancer cells with a maximum effect at 50 µM and showed a lower toxicity in MDCK cells. Long-term efficacy (20 days) was evidenced by CBD-Nem at inhibiting colony formation in cancer cells. Furthermore, CBD-Nem reduced the proportion of cells in the G2-M phase, induced apoptosis, and inhibited the migration and invasion of CF41.Mg cells. **Conclusions**: CBD-Nem exhibited an in vitro antitumor effect, which supports its study in dogs with mammary carcinoma.

## 1. Introduction

Mammary tissue can undergo cell transformation due to genetic and epigenetic factors, leading to tumor development, one of the most common pathologies in unsterilized female dogs and women [1]. Mammary tumors in female dogs show many similarities with the disease in women [2]. Between 25 to 70% of unsterilized female dogs will have a mammary tumor during their lifetime, and approximately 50–60% of these will be of malignant pathological nature [3,4]. Approximately 90% of these tumors correspond to carcinomas of different histological types [5]. The treatment of choice is surgery, and the post-procedure survival rate is variable; local recurrence and metastasis are usual [6]. Other therapeutic alternatives, especially in cases of high malignancy, include chemotherapy. Chemotherapy is non-selective and usually results in severe adverse effects [7]. Consequently, there is a need to search for new strategies with therapeutic potential for this highly recurrent disease.

The cannabis plant has wide medicinal properties, as it contains more than 100 phytocannabinoids. The two most abundant and studied phytocannabinoids are tetrahydrocannabinol (THC), which has psychotropic effects, and cannabidiol (CBD), which lacks psychotropic properties. CBD has shown the capacity to inhibit cell proliferation and migration, inducing apoptosis in various tumor cells, including human mammary cells [8,9]. There are few data concerning the effect of CBD on canine mammary cancer cells [10]. The described anticancer effects of CBD are mediated through interaction with different membrane receptors, such as CB1, CB2, 5HT1A, GPR55, and TRPV1-4. In addition, intracellular receptors such as PPARs and VDAC1 have also been involved [9,11,12]. With these receptors, CBD can modulate different signaling pathways such as MAPK/ERK, mTOR-Akt, and COX-2-BCL2, among others [13].

Triple-negative breast cancer (TNBC) is a subgroup of mammary tumors of high malignancy, and few treatment options are available for this disease [14]. CBD may play a therapeutic role in the treatment of TNBC. It has been reported that CBD inhibits TNBC cell proliferation, migration, and invasion by suppressing the activation of the EGF/EGFR pathway and its downstream targets (Akt and NF-κB), as well as suppressing MMP2 and MMP9 secretion [8].

CBD belongs to class II of the Biopharmaceutical Classification System (BCS) [15], whose components show low water solubility and high permeability. Thus, their absorption is limited by the dissolution rate. CBD has low solubility, with an aqueous solubility of only 0.01 μg/mL [16,17], which contributes to its limited biological effects [18]. The investigations/therapeutic evaluation with CBD is usually limited and requires organic solvents such as ethanol, DMSO, or dispersing agents (for example, polyoxyethylene glycerol triricinoleate or poloxamer 188); those solvents/agents can be used in low concentrations limits the dose that needs to be administered [19]. A technological approach to improve the aqueous compatibility of lipophilic compounds, such as CBD, in addition to protecting the molecule from physiological and environmental conditions, is the incorporation into nanocarriers.

Various lipid-based nanocarriers have been described for the delivery of CBD, including nanoliposomes, Nano emulsions, nanostructured lipid carriers (NLCs), and solid lipid nanoparticles (SLNs). However, to date, there is limited information regarding the antitumor efficacy of CBD loaded into these nanocarriers. The effect of CBD in NLCs has been evaluated in a human glioblastoma cell line (U373MG), showing a reduction in cell viability [20]. On the other hand, an in vitro study analyzed the antineoplastic activity of CBD encapsulated in poly (lactic-co-glycolic acid) (PLGA) nanoparticles in human ovarian cancer cell lines (SKOV-3, OAW-42, IGROV-1), demonstrating effectiveness in decreasing cell viability [18]. Another study incorporated CBD into nanomicelles and analyzed its effect on different breast cancer cell lines (MDA-MB-231, 4T1, and MCF-7), observing a reduction in cell viability [21]. To date, no studies have evaluated the effect of CBD-loaded nanocarriers on canine cancer cells.

Oil-in-water nanoemulsions (Nem) are heterogeneous systems consisting of an oil phase dispersed in an aqueous medium and stabilized by surfactants [22]. This approach provides a potential delivery strategy for CBD, avoiding the use of toxic solvents/agents [23]. It has been demonstrated that curcumin, a highly lipophilic molecule like CBD, when vehiculized in Nem, prolongs the anticancer effect on cell cultures and in vivo models [24]. However, to date, no studies using CBD-loaded nanoemulsions have evaluated their antineoplastic potential in human or animal cancer cells. Currently, the effects of CBD dissolved in ethanol or dimethylsulfoxide (DMSO) have been tested on various lines of human breast cancer cells, including ER-positive cells (MCF-7, ZR-75-1, T47D), ER-negative cells (MDA-MB-231, MDA-MB-468, and SK-BR3), and triple-negative cells (SUM159, 4T1up, MVT-1, and SCP2), as well as on a Luminal B-type canine mammary carcinoma cell line (CMT12) [9,10]. The present study aimed to develop CBD loaded in Nem (CBD-Nem) to assess in vitro safety (MDCK cell line) and antitumor activity on canine (CF41.Mg, IPC366 cell lines) and human (MDA-MB-231, used as control) mammary carcinoma cells. Moreover, uptake, internalization, apoptosis, colony formation, migration, and invasion in mammary canine cancer cells in response to CBD-Nem were evaluated.

## 2. Materials and Methods

### 2.1. Drugs

Isolated powdered CBD (98%, Clever Leaves, Bogotá, Colombia) was used. Concentrations of 10, 20, 30, 40, and 50 µM of CBD were loaded into Nem (CBD-Nem, or dissolved in ethanol (CBD-E), and tested based on effective doses previously reported in human breast cancer cells, including ER-positive cells (MCF-7, ZR-75-1, T47D), ER-negative cells (MDA-MB-231, MDA-MB-468, and SK-BR3), and triple-negative breast cancer (TNBC) cells (SUM159, 4T1up, MVT-1, and SCP2) [9,25,26]. Rhodamine 6 G (Rho) (Sigma-Aldrich, St. Louis, MO, USA) was used to study uptake and internalization provided by Nem in the CF41.Mg cell line.

### 2.2. Formulation of CBD-Nem and Rho-Nem

Nem was prepared at room temperature using the solvent displacement method previously described for Guerrero et al., 2018 [24]. Briefly, an organic phase containing 125 µL of oil (triglyceride ester of saturated caprylic and capric fatty acids, Miglyol 812, Sasol GmbH-Germany, Hamburg, Germany) (2.5% *v*/*v* in the final volume), 30 mg of a surfactant (soybean lecithin fraction enriched with phosphatidylcholine, Epikuron 145 V, Cargill-Spain, Barcelona, Spain) (0.6% *w*/*v* in the final volume), and 10 mL of ethanol (Merck-Germany, Darmstadt, Germany) was added to an aqueous phase containing Milli-Q water (20 mL). The sample was concentrated from 20 mL to 5 mL to eliminate organic solvents (ethanol) and to concentrate the components using rotary evaporation, with the volume measured periodically using a graduated cylinder until the final volume was reached [24]. CBD-Nem was prepared in a similar manner, but completely dissolving 25 mg of CBD (0.1% *w*/*v* in the final volume) in a Miglyol–ethanol–Epikuron mixture. For uptake and internalization assays, a fluorescent lipophilic component, Rho (MW: 479 g/mol, LogP 6.3, Sigma-Aldrich, St. Louis, MO, USA), was incorporated into the Nem (Rho-Nem) instead of CBD (MW: 314.47; LogP: 6.3) [27]. This procedure was performed under the same conditions described for CBD-Nem.

### 2.3. Physicochemical Characterization of CBD-Nem and Rho-Nem

Hydrodynamic diameter (including polydispersity index/PDI) and zeta potential were determined by dynamic light scattering (DLS) and laser Doppler anemometry (LDA), respectively, using a ZetaSizer NanoZS (Malvern Instruments, Malvern, UK) equipped with a standard laser (λ = 633 nm) as an incident beam. The formulations were diluted 20 times in Milli-Q water and loaded into a disposable folded capillary cell (DTS1070). Each analysis was performed in triplicate at 25 °C. Particle concentration (number of particles per mL) was determined by nanoparticle tracking analysis (NTA) using a NanoSight NS300 (Malvern Instruments, Malvern, UK). The samples were diluted 10–20 times with Milli-Q water to achieve an optimum concentration range of 10^9^ to 10^11^ particles/mL. A minimum of five videos (one minute each) of the particles moving under Brownian motion were captured [28]. Data were expressed as mean ± standard error from three independent experiments.

To evaluate the stability of CBD-Nem over time, the particle size and zeta potential were analyzed immediately after formulation (day 1). Three samples prepared on three different days were stored at 4 °C, and the same parameters were measured after 4, 10, 30, and 215 days. Data were expressed as mean ± standard error.

### 2.4. Morphological Characterization of CBD-Nem

Morphological characterization was carried out using a scanning transmission electron microscope (STEM), model Inspect F-50 (FEI, The Netherlands). STEM images were obtained by placing a drop (20 µL) of the formulation on a copper grid (200-mesh, coated with Formvar) for 2 min. The drop was then removed with filter paper without touching the grid, and the grid was washed twice with a drop of Milli-Q water for 1 min, removing the drop with filter paper. A drop of phosphotungstic acid (1% *w*/*v*) was added to the grid for 2 min and then removed with filter paper. Finally, the grid was air-dried at room temperature for at least 1 h before visualization [28,29].

### 2.5. Encapsulation Efficiency in CBD-Nem

The encapsulation efficiency (AE%) of CBD in CBD-Nem was determined by analyzing the difference between the total amount of CBD added to the formulation and the non-encapsulated/free CBD (AE% = (amount of CBD encapsulated in the formulation/total added CBD) × 100). The amount of CBD encapsulated was estimated indirectly by analyzing the non-encapsulated CBD and measuring absorbance at 276 nm using a UV spectrophotometer (Cary Eclipse, Agilent, Santa Clara, CA, USA). The non-encapsulated CBD was quantified by isolating an aliquot of CBD-Nem using Vivaspin^®^ tubes (MWCO 3000 Da, 4000 RCF × 25 min) and then dissolving the ultrafiltrate in ethanol (1:3 *v*/*v*) for the absorbance measurements [27]. A calibration curve of absorbance was performed with CBD dissolved in water/ethanol (1:3 *v*/*v*) at different concentrations, resulting in a coefficient of determination (R^2^) of 0.993 (see Appendix A). A detection limit of 0.05 mg/mL was obtained. The quantification limit was estimated at 0.08 mg/mL. The mean ± standard error from three independent experiments was considered to obtain AE%.

### 2.6. CBD-Release from CBD-Nem

CBD release studies from CBD-Nem were conducted by incubating a sample of 1 mL of the formulation in a dialysis bag (pore size of 10,000 Da), simulating a physiological environment with phosphate-buffered saline (PBS) pH 7.4 or an acidic medium with PBS pH 5.5 (polysorbate 80, 0.5% *w*/*v* was added to ensure proper sink conditions due to the limited solubility of CBD in the aqueous medium) [23,29]. The dialysis bags were placed in an incubator at 37 °C for horizontal agitation. At different times (0, 1, 2, 4, 8, 12, 48, 72, 144, 168, 240, 312, 336, 384 h), a total of 1 mL of the solution was collected and diluted in ethanol (1:3 *v*/*v*) to analyze CBD content by UV spectrophotometry (276 nm). From a kinetic perspective, the cumulative CBD release data in media at pH 5.5 and 7.4 were fitted to different kinetic models using the DDSolver add-in for Microsoft Excel. Model selection was based on the adjusted coefficient of determination (R^2^). A calibration curve performed with CBD dissolved in PBS + polysorbate 80/ethanol (1:3 *v*/*v*) was elaborated, and a coefficient of determination (R^2^) of 0.992 for pH 5.5 and 0.991 for pH 7.4 (see Appendix A) was obtained. The detection limit was 0.05 mg/mL in PBS medium at pH 5.5 and 0.06 mg/mL in medium at pH 7.4. The quantification limit was estimated at 0.08 mg/mL in both media. Data were expressed as mean ± standard error from three independent experiments.

### 2.7. Cell Culture

The selected neoplastic cell lines were canine triple-negative mammary carcinoma CF41.Mg (ATCC, catalogue number CRL-6232) and IPC-366 (Cellosaurus, Swiss Institute of Bioinformatics, catalogue number CVCL_0Q77) (kindly provided by Prof. J.C. Illera, Faculty of Veterinary Medicine, Complutense University of Madrid). The IPC-366 cell line was validated and characterized by Caceres et al., 2015 [30] and represents inflammatory mammary carcinoma. Both cell lines were maintained in high-glucose Dulbecco’s Modified Eagle Medium (DMEM) supplemented with 10% fetal bovine serum (FBS), 2 mM glutamine, penicillin G, streptomycin, and amphotericin B. The non-neoplastic cell line, derived from canine renal tubular epithelium, MDCK (ATCC, catalogue number NBL-2), was cultured in minimum essential medium (MEM) supplemented with 10% FBS and antibiotics. Additionally, a human TNBC cell line, MDA-MB-231 (ATCC, HTB-26), was cultured in high-glucose DMEM supplemented with 10% FBS, 2 mM glutamine, 1% pyruvate, and antibiotics. All cultures were maintained at 37 °C in a humidified atmosphere with 5% CO_2_, and the culture medium was replaced every 48 h after washing the cells with Dulbecco’s phosphate buffered saline (DPBS). For cell dissociation, once 70–80% confluence was reached, the cells were washed with DPBS buffer and incubated with 2.5% trypsin/ethylenediaminetetraacetic acid (EDTA). The reaction was stopped with high-glucose DMEM supplemented with 10% FBS. The cells were then centrifuged at 200× *g* for 10 min, and the pellet was recovered, counted, and reseeded. Cell concentration was assessed using the trypan blue exclusion method and hemocytometry [31].

### 2.8. Flow Cytometry (FACS) Assays

CF41.Mg cells were seeded in flasks at a density of 2 × 10^5^ cells per well and incubated for 24 h. Subsequently, the cells were exposed to Rho-Nem, Rho in ethanol (Rho-E) at 1 μM, or vehicle control at 5% (*v*/*v*) for an additional 24 h. Afterwards, the cells were collected in borosilicate glass tubes, centrifuged, resuspended in FACS buffer (DPBS containing 1% FBS and 5 mM EDTA), and evaluated by FACS (Canto A, Becton Dickinson, Franklin Lakes, NJ, USA) [27]. Mean fluorescence intensity relative to controls (MFRC) of the cell populations from the different treatment groups was determined and reported as the mean ± standard deviation from three independent experiments. FlowJo software version 7.6 was used for the analysis of the data.

### 2.9. Confocal Fluorescence Microscopy

The intracellular distribution was evaluated using fluorescence microscopy. The 5 × 10^4^ CF41.Mg cells were seeded in 12-well plates on sterile coverslips (18 mm round coverslips, Deckglässer vessels) and cultured for 24 h in DMEM medium. The cells were treated with Rho-Nem 1 µM at 5% (*v*/*v*) (Nem, 1 µM Rho-E, and ethanol used as controls). The samples were incubated 24 h at 37 °C. The cells were washed with PBS at 4 °C and fixed with paraformaldehyde (4%) for 15 min. The coverslips were mounted on glass slides using a fluorescence mounting medium containing 4′,6-diamidino-2-phenylindole (DAPI, blue staining) to counterstain the nuclei. Confocal images were acquired using a Zeiss LSM700 microscope (Zeiss, Hebron, KY, USA). The images were analyzed using ImageJ software version 1.50 [27]. Three independent experiments were performed.

### 2.10. Cell Viability Assay

CF41.Mg, MDCK, IPC366, and MDA-MB-23 cells were seeded (1.5 × 10^3^ cells per well) in a 96-well plate. For the CF41.Mg and MDCK cells, after 24 h, different concentrations of CBD-Nem or CBD-E at 5% (*v*/*v*) (0, 10, 20, 30, 40, 50 µM) were added, using pertinent controls to compare (control medium, ethanol, and Nem). The cells were incubated for 24 and 48 h with CBD-E and for 24, 48, and 72 h with CBD-Nem. For the IPC-366 and MDA-MB-231 cells, treatments of CBD-Nem were administered at concentrations of 20 µM and 50 µM (5% *v*/*v*). Cell viability was assessed using the 3-(4,5-dimethylthiazol-2-yl)-5-(3-carboxymethoxyphenyl)-2-(4-sulfophenil)-2H-tetrazolium (MTS) reduction assay (CellTiter 96^®^ AQueous One Solution; Promega Corporation, Madison, WI, USA), determining the absorbance at 490 nm with a microplate reader [32]. Each experiment was conducted in triplicate with at least three independent repetitions.

### 2.11. Clonogenic Assay

A total of 500 CF41.Mg cells were seeded, in duplicate, in 6-well plates with 1 mL DMEM medium. After 2 h to allow for cell attachment, treatments were applied. These treatments included control medium, ethanol, Nem, CBD-E 20 µM, and CBD-Nem at 20 and 50 µM (all treatments were added at 5% *v*/*v*). The cells were incubated for 20 days to allow colony formation. Once colony formation was confirmed, the medium was removed, and the cells were washed with PBS and fixed with 0.5% crystal violet dissolved in methanol/water for 45 min. After fixation, the wells were rinsed with water, and individual colonies of at least 50 cells were counted in at least 10 random fields under 4× magnification microscopy [33]. ImageJ software with the “cell counter” tool was used. Each experiment was conducted in triplicate with at least three independent repetitions.

### 2.12. Cell Cycle Assay

CF41.Mg cells (3 × 10^5^) were seeded in duplicate. After 24 h, cell cycle synchronization was performed by serum starvation for an additional 24 h. The cells were then washed with DPBS, and the medium was replaced to expose the carcinoma cells to experimental conditions (CBD-E 20 µM, CBD-Nem 20 and 50 µM) along with their respective controls, applied at 5% (*v*/*v*). After 24 h of incubation, the cells were detached using trypsin/EDTA following a DPBS wash. The supernatant was retained, and the number of viable cells per flask was determined. The cells were centrifuged, the supernatant was discarded, and the cell pellet was fixed with cold methanol (−20 °C) and incubated for 30 min at −20 °C. Afterwards, the cells were centrifuged again, and the supernatant was removed. Pellets were washed twice with DPBS, and RNase was directly added (10 µL, 100 µg/mL) to the cell pellets. Finally, 10 µL of propidium iodide was added, and samples were analyzed by FACS (Canto A, Becton Dickinson). Three independent experiments were conducted and analyzed using ANOVA or the Kruskal–Wallis test.

### 2.13. Apoptosis Assay

An annexin V/propidium iodide (PI) assay (Thermo Fisher Scientific, Inc., Waltham, MA, USA) was performed. CF41.Mg cells (2 × 10^5^) were seeded in duplicate flasks. After 24 h, the cells were washed with DPBS and the medium was incubated with CBD-E 20 µM, CBD-Nem 20 µM, and 50 µM, along with their respective negative controls. Doxorubicin (1 µM) was used as a positive control for apoptosis induction. All treatments were applied at 5% (*v*/*v*). After 24 h, the cells were detached and counted. The cells were centrifuged, and the cell pellet was resuspended in annexin binding buffer. Then, 5 µL of Alexa Fluor-conjugated annexin V was added, followed by 1 h of incubation. Subsequently, 10 µL of PI was added. Finally, the cell suspensions were analyzed using FACS (Canto A, Becton Dickinson). Four independent experiments were performed [32].

### 2.14. Cell Migration and Invasion Assays

Cell migration was assessed using 24-well Transwell^®^ cell culture inserts with an 8 µm pore size (BD Biosciences). A total of 5 × 10^4^ CF41.Mg cells were seeded in duplicate in the upper chamber and incubated for 6 h in the presence of CBD-E 20 µM, CBD-Nem 20 µM, and 50 µM, and all treatments were applied at 5% *v*/*v*, as well as the respective controls. A 5% FBS gradient was used in the lower chamber. Non-migrating cells on the upper surface were removed using a cotton swab, and the membrane was fixed with cold methanol for 20 min and stained with 1% Giemsa. The polycarbonate membrane was then detached with a scalpel and mounted on a microscope slide. Randomly selected fields (6 per membrane at 10× magnification) were analyzed to count migrating cells using ImageJ software with the “cell counter” tool [31]. In addition, cell invasion experiments were performed using Transwells under the same conditions described for migration, except that the inserts were pre-coated with Matrigel (BD Biosciences, Franklin Lakes, NJ, USA). CF41.Mg cells were incubated for 24 h [25]. The experiments included three independent trials.

### 2.15. Data Analysis

DLS results were analyzed using the ZetaSizer software v7.12; each analysis was performed in triplicate at 25 °C. NTA data were analyzed with the built-in NTA v 3.0 software (Malvern, UK) [23]. The Shapiro–Wilk test was employed to determine the data distribution type. ANOVA with a Tukey post hoc analysis or Kruskal–Wallis test was performed to determine differences between control and treatment groups. Repeated measures of ANOVA were used to measure CBD release. IC_50_ values were obtained by fitting the data to a 4-parameter logistic regression. Statistical significance was considered at *p* < 0.05. These analyses were conducted using GraphPad Prism^®^, DDSolver, and Excel software.

## 3. Results

### 3.1. Physicochemical Characterization of the Formulations

CBD-Nem and Nem were developed using the solvent displacement methodology. In general, the characteristics of the nanoemulsions were similar (Table 1), ranging between 140–170 nm, with polydispersity (PDI) of 0.1–0.14 and negative zeta potentials (−43–−61 mV). NTA revealed the presence of nanostructures ranging from 7.4^12^ to 1.2^14^ nanoparticles per mL. STEM analysis for CBD-Nem showed spheroidal nanostructures (Figure 1A), the same size as shown by DLS (Table 1). The stability test showed that CBD-Nem maintained nanometric size values at 7 months (147.1 ± 0.05 nm), similar to the initial measurements. After this period, they also retained zeta potentials greater than −30 mV (−39.2 ± 0.9 mV), indicating that no nanoemulsion aggregation occurred during storage [34] (Figure 1B).

### 3.2. CBD Encapsulation Efficiency and Drug Release

As shown in Table 1, the CBD encapsulation efficiency (AE%) in the CBD-Nem formulation was almost quantitative (≥99%, equivalent to 4.9 mg/mL of encapsulated CBD), demonstrating the effectiveness of the selected nanoemulsification strategy to entrap the lipophilic drug. Following the incorporation of CBD into the Nem, the drug release profile was determined. Additionally, considering the intended topical application of the formulation to the affected area following surgical removal of the primary tumors (to prevent/treat recurrent tumor growth and metastasis of remaining cancer cells due to the high aggressiveness of TNBC) [14], and to ensure high concentrations of the CBD-Nem formulation in the tissues, we decided to evaluate CBD release over a period longer than 30 days to estimate its potential application in future clinical studies. A physiological environment was simulated with PBS at pH 7.4, and a tumor microenvironment with PBS at pH 5.5 at 37 °C. The cumulative release profile of CBD from the Nem is shown in Figure 2. A slow drug release percentage was observed, starting on day 3 at pH 5.5 and on day 6 at pH 7.4. The cumulative release at pH 5.5 was 8 ± 2, 21 ± 17, 40 ± 12, 72 ± 3, and 92 ± 2% on days 3, 16, 25, 34, and 36, respectively. At pH 7.4, the cumulative release was 1 ± 1, 15 ± 1, 52 ± 1, and 79 ± 1% on days 6, 18, 29, and 36, respectively.

### 3.3. Rho-Nem Uptake and Internalization

For Rho-Nem uptake and internalization, the CF41.Mg cells were selected because they are a highly aggressive triple-negative mammary carcinoma. Rho was chosen due to its fluorescent characteristics and similar lipophilic nature and molecular weight as CBD (CBD has no fluorescent properties). Rho-Nem reached a size and zeta potential similar to CBD-Nem (Table 1). FACS assays indicated that Rho-Nem was efficiently taken up and maintained by the cancer cells for 24 h. Rho fluorescent levels were significantly higher (*p* < 0.0001) for Rho-Nem (14 ± 1 MFRC) compared to Rho-E (11 ± 1 MFRC) (Figure 3A). Confocal microscopy experiments were performed to identify the intracellular distribution of the lipophilic payload. As shown in Figure 3B, an increased fluorescence was observed with Rho-Nem compared to Rho-E. A homogeneous distribution throughout the cytoplasm of CF41.Mg cells after 24 h are shown.

### 3.4. Viability Assays in Canine Mammary Carcinoma and Canine Renal Epithelium Cell Lines

To estimate the effect of CBD-Nem on the viability of mammary cancer cells, an MTS reduction assay was performed. CBD-Nem induced a concentration-dependent decrease in CF41.Mg cell viability at 48 h (*p* < 0.0001) and 72 h (*p* < 0.0001), an effect that began at 30 µM and became more significant at 50 µM (*p* < 0.0001) (Figure 4A). CBD-E induced a significant reduction in cell viability (*p* < 0.0001) starting at 20 µM, at both 24 and 48 h (*p* < 0.0001) (Figure 4B).

To determine whether the behavior of CBD-Nem is similar in other types of mammary cancer cell lines, the effect of CBD-Nem was evaluated on IPC366 canine inflammatory carcinoma cells, which represent a highly aggressive type of carcinoma in female dogs. The human MDA-MB-231 cell line, for which there is already information regarding the effect of CBD delivered in ethanol and DMSO [13], was also studied. In IPC366 cells, CBD-Nem induced a significant decrease in cell viability (*p* < 0.0001) at a concentration of 20 µM, resulting in reductions in viability at both 48 h (*p* = 0.0006) and 72 h (*p* = 0.0028). At 50 µM, after 72 h, CBD-Nem more efficiently reduced the cell viability (*p* < 0.0001) compared to 20 µM (*p* = 0.0028), indicating a concentration-dependent effect (Figure 5A). CBD-E at 20 µM also caused a significant reduction in cell viability (*p* < 0.0001) at both time points analyzed. Additionally, MDA-MB-231 cells also exhibited sensitivity to CBD-Nem, with a reduction in cell viability that was similarly significant as CBD-E, starting from 20 µM (*p* < 0.0001) at 48 and 72 h. In these cells, a concentration-dependent effect of CBD-Nem was also observed at 50 µM (*p* < 0.0001) at both 48 and 72 h (Figure 5B).

In the non-neoplastic MDCK cells, treatment with CBD-Nem induced a significant reduction in cell viability (*p* < 0.0001) starting at 30 µM and with a greater decrease in cell viability at 24 h (Figure 6A). It was not possible to calculate the IC_50_ at these concentrations as viability decreased by less than 50%. CBD-E significantly decreased cell viability in a dose-dependent manner (*p* ≤ 0.0001), an effect observed with 20 µM at 24 and 48 h (Figure 6B), showing greater toxicity than CBD-Nem. The IC_50_ values for MDCK were 20 ± 0.2 µM and 25 ± 2 µM at 24 and 48 h, respectively.

These results show that CBD-Nem triggered different effects in carcinoma cells than in non-neoplastic cells, with a time-dependent reduction observed in the CF41.Mg line, exhibiting a greater reduction in cell viability at 72 h (43 ± 11%) compared to 24 h (17 ± 7%) (*p* < 0.0001). In contrast, in the MDCK line, a time-inverse reduction in cell viability was observed, with a greater reduction at 24 h (37 ± 11%) than at 72 h (17 ± 11%). Significant differences (*p* ≤ 0.0001) between CF41.Mg and MDCK cells at 72 h with 50 µM CBD-Nem were observed, with a greater decrease in carcinoma cells (Figure 6C).

### 3.5. Colony Formation Assay

To evaluate the effect of CBD-Nem on the proliferation of CF41.Mg cells over prolonged periods, and mimicking a drug treatment with prolonged effect, a colony-forming (clonogenic) assay was performed for 20 days. Cell colony counts revealed that CBD-Nem 50 µM and CBD-E 20 µM completely inhibited the colony-formation; in addition, a significant reduction in the number of colonies (from 38 to 11 colonies) was noted when cells were exposed to CBD-Nem at 20 µM (Figure 7). Below the graph, the visual appearance of a representative experiment applying the treatments in comparison with relevant controls can be appreciated.

### 3.6. Cell Cycle and Apoptosis Assays

For the cell cycle assay, propidium iodide (PI) staining was performed to identify changes in the proportion of cells in different phases of the cell cycle in response to CBD-Nem and relevant controls. It was observed that CBD-Nem at 20 µM and 50 µM significantly reduced G2-M phase entry in CF41.Mg cells (Figure 8A). CBD-E at 20 µM also decreased G2-M phase entry in these carcinoma cells, although to a lesser extent. In addition, a significant increase in the S phase was observed in CF41.Mg cells treated with CBD-Nem 50 µM. No significant differences were found in the proportion of CF41.Mg cells in the G0–G1 phase between treatments and their respective controls. Regarding apoptosis, annexin V/PI was selected as staining and doxorubicin was added as positive control. A significant increase in apoptosis was observed in CF41.Mg cells treated with 20 µM CBD-E, and 20 and 50 µM CBD-Nem, comparable to the increase induced by the positive control, doxorubicin (Figure 8B). Apoptosis was increased by 3 ± 0.3-fold with 20 µM CBD-Nem and by 2.8 ± 0.1-fold with 50 µM CBD-Nem, compared to the untreated control (no significant differences found between the two treatments).

### 3.7. Cell Migration and Invasion Assays

Migration and invasion assays were conducted with CF41.Mg cells in the presence of CBD-Nem 20 and 50 µM, and CBD-E 20 µM. It was observed that both CBD-Nem and CBD-E significantly reduced cell migration (Figure 9A). CBD-Nem 50 µM and CBD-E 20 µM resulted in a more pronounced reduction in migration than 20 µM CBD-Nem. Nevertheless, all three treatments exhibited the same level of statistical significance compared to the medium control. In the cell invasion assay, all three treatments (CBD-Nem 20 and 50 µM, and CBD-E 20 µM) also showed a significant reduction compared to controls (Figure 9B). Similarly, for migration, the reduction was more pronounced in the presence of CBD-Nem 50 µM and CBD-E 20 µM; no significant differences were detected between these treatments.

## 4. Discussion

In mammary cancer, high-grade tumors are often treated with chemotherapy as an adjuvant and palliative strategy. However, this therapeutic alternative is not selective and can induce several side/adverse effects. CBD, a compound with demonstrated antineoplastic capacity against various malignant tumors [9,26], may represent an alternative. However, applications of CBD are limited due to low solubility in biological media, instability, and poor cellular uptake and internalization [35,36]. These CBD characteristics also contribute to the variability in experimental results reported in the literature [37]. In this study, we prepared a nanoemulsion previously formulated by our team [24], which was able to solubilize CBD (in the oil nucleus of the nanoemulsion), allowing the dispersibility of CBD in biological media, protecting the molecule from environmental adversities, and increasing the cellular uptake and internalization. The oil used (Miglyol 812, Sasol GmbH-Germany, Hamburg, Germany) corresponds to a medium-chain triglyceride (MCT) where CBD shows high solubility (273.8 mg/mL), which allowed a complete dissolution of the drug [17]. In addition, CBD-Nem was developed avoiding the use of toxic solvents and employing FDA-approved excipients, making it suitable for in vivo applications in preclinical and clinical protocols. Unlike other strategies used to create CBD-Nem, such as those that employ sonication and two-step high-pressure homogenization [38], the methodology described here (solvent displacement) is less polluting, consumes less energy, and is more easily scalable.

CBD-Nem had a spherical shape (Figure 1A), size in the range of ~150 nm, with polydispersity ≤ 0.15, and zeta potential of ~−50 mV (Table 1). These values are ideal since they contribute to the stability of the nanosuspension in biological medium, preventing self-aggregation [39]. These results resemble those found in other CBD-loaded nanoemulsions, as described by Lewinska (2021) [38] and Fathordoobady et al., 2021 [40], where zeta potential values around −50 mV were achieved, using surfactin obtained from *Bacillus subtilis* natto (KB1) and lecithin as surfactants. Unlike our formulation, the referred studies utilized high-energy techniques (sonication, high-pressure homogenization, ultrasonication, and microfluidization) to produce CBD-loaded nanoemulsions. In addition, as evidenced in Figure 1B, the proposed nanoformulation in this paper is highly stable in aqueous medium. Furthermore, the concentration of nanocarriers (nanoparticles per mL) obtained in this work is in the superior range (~1–5 × 10^13^) as presented in other works developing nanoparticles [27,28,41,42], representing an efficient methodology for industrial purposes.

CBD-Nem exhibited very high encapsulation efficiency (99%, Table 1), which is equivalent to 5 mg/mL. Other formulations, such as poly (lactic-co-glycolic acid) (PLGA) nanocapsules loaded with CBD, have reached encapsulation efficiencies around 95% and encapsulating between 0.1 to 0.8 mg CBD/10 mg PLGA [23].

The release kinetics of CBD from Nem in PBS were tested at pH 5.5 and 7.4, with no detectable concentrations until the third and sixth day, respectively (Figure 2). However, this lack of detection could be related to the detection limit of CBD obtained through UV spectroscopy, which was 0.05 mg/mL. In this regard, the use of chromatographic techniques with greater analytical sensitivity could improve detection thresholds [43]. The release assay described here showed a slow release, slightly faster at pH 5.5, but without significant differences until day 34. The release model with the highest fit, according to the observed determination coefficient (R^2^ = 0.9), was the Hopfenberg kinetics model, indicating that the main limiting process in the release of CBD from nanoemulsions was matrix erosion [44]. The slow release of CBD from nanoemulsions could be related to the high lipophilicity of the CBD molecule, which results in a greater affinity for the oily core than for the external medium [45]. Additionally, particle size may also play a role, as described by Aparicio-Blanco et al., 2018 [20], who reported slower CBD release from 50 nm PLGA nanocapsules compared to 20 nm ones. PLGA microcapsules of 24 µm have also shown sustained release, with 80% of CBD released after 42 days [18]. Other studies using CBD in Nem and manufactured using microfluidic and ultrasound energies (using lecithin as emulsifier and poloxamer 188 as co-emulsifier) yielded significantly faster drug release profiles. Using simulated gastric fluid (SGF, pH = 1.3) and simulated intestinal fluid (SIF, pH = 7.0), abrupt release rates were observed within the first hour of exposure at both pH levels, followed by a reduced release rate during the subsequent 3 and 12 h, respectively, with a more pronounced effect in SGF [40]. Duse et al., 2019 [46] reported that the effect of pH on lipophilic drug release depends on the erosion contribution as the release mechanism, describing faster release from PLGA nanoparticles at pH 5.5 compared to 7.4.

Rho was used to indirectly assess CBD internalization in CF41.Mg cells due to physicochemical similarities with CBD (Rho: molecular weight of 479 g/mol and a LogP of 6.3; CBD: molecular weight of 315 and LogP: 6.3). The results of FACS showed a more efficient uptake when encapsulated in Nem compared to the ethanol vehicle (Figure 3A). Rho-Nem achieved 29% higher fluorescence in CF41.Mg cells than Rho-E. These results were further supported by a confocal microscopy evaluation, showing higher intracellular fluorescence in CF41.Mg cells treated with Rho-Nem compared to those treated with Rho-E (Figure 3B). This finding agrees with a previous study using the lipophilic and fluorescent curcumin loaded in Nem, which reported a significant increase in uptake (22%) and internalization (evidenced by a stronger intracellular fluorescent signal of curcumin) than curcumin dissolved in DMSO, demonstrating enhanced endosomal uptake [24]. However, this study represents a preliminary and indirect evaluation of CBD uptake and internalization. Since Rho is a charged molecule (like curcumin), unlike CBD, and is not structurally similar to CBD, the mechanisms of uptake and internalization may differ. Facilitated transport or endocytosis may play a more prominent role in Rho internalization, whereas CBD could also enter cells through passive diffusion [47]. Previous studies have shown that a negative surface charge on nanocarriers (−30 to −50 mV) does not hinder cellular internalization [24,27,48,49,50]. In addition, the nanoscale size of these carriers (100–250 nm) facilitates internalization [24,51,52]. Furthermore, the intrinsic permeation properties of the phospholipids used for stabilization in this paper (surfactant of soybean lecithin) also favor the internalization of Nem [27].

To date, nanoemulsions containing CBD have not been evaluated on canine mammary cancer cells. In this work, the canine cell line CF41.Mg and IPC366 were selected. In addition, the human mammary cancer cell line MDA-MB-231 was also tested to compare the results and show the potential for this formulation in human applications. The cell line MDCK (canine kidney cells) was used as a control of non-cancer cells. Cell viability was assessed by seeding all cell lines at the same density (1.5 × 10^3^) to ensure reproducible and comparable results. We observed that CBD-Nem reduced the viability of mammary carcinoma cells, effects that were time- and concentration-dependent. The reduced response of CF41.Mg cells to CBD-Nem (Figure 4A) compared to the ethanol vehicle (Figure 4B). Additionally, CF41.Mg cells exhibited lower susceptibility to the effects of CBD-Nem compared to IPC366 and MDA-MB-231 cells, where a greater decrease in viability was observed at 20 µM (Figure 5A,B), although these effects were less pronounced than those of CBD-E in both cell lines. Nevertheless, it is relevant that the cytotoxic effect of CBD-Nem is reproducible in different types of canine and human mammary tumor cells representative of high-grade tumors. The less pronounced effect of CBD-Nem compared to CBD-E on the cell viability of the evaluated cancer cell lines, without achieving an IC_50_ even at the highest tested dose (50 µM), may be attributed to several factors. One of them is the slow release of CBD from the nanoemulsions (Figure 2), which could hinder, within the evaluated time frame, the interaction of CBD with membrane receptors such as GPR55 and TRPVs, which can modulate the viability of cancer cells. Moreover, the nanoemulsion may be fully internalized by cancer cells, as nanocarriers smaller than 500 nm—both cationic and anionic—are internalized via pinocytosis mechanisms [47]. This could delay the interaction of CBD with intracellular receptors until it is released from the nanoemulsion. The nanoemulsion would enhance CBD internalization into cancer cells, as suggested by outcomes with Rho-Nem. In this regard, we hypothesize that increased internalization may enhance interactions of CBD with intracellular targets such as PPARγ (involved in the induction of apoptosis), mitochondrial receptors like the voltage-dependent anion channel 1 (VDAC1), and endoplasmic reticulum calcium channels [9]. In contrast, ethanol as a vehicle may favor CBD interactions with membrane receptors such as TRPV, GPR55, and CB1-2 [53]. For this reason, the nanoemulsion could show better results at longer time points, as observed in the colony formation assays.

Interestingly, CBD-Nem triggers different effects on the viability of CF41.Mg (Figure 4A) compared to MDCK cells (Figure 6). Cytotoxic effects were observed in carcinoma cells over time, while non-tumor cells showed recovery of viability at 72 h (Figure 6C), indicating a protective effect (related to safety) for the CBD-Nem formulation. This differential effect could be explained by differences in the CB1-CB2 receptor expression in both cell lines. MDCK cells express higher levels of CB1 than CB2 receptors [54], and CF41.Mg cells have higher CB2 expression than CB1. In human breast cancer, CBD has been reported to inhibit cell growth and induce apoptosis through the activation of both CB2 and TRPV1 [55]. CB1 activation in renal tissue is associated with Na+/K+ ATPase pump activation and mitochondrial fission stimulation [56,57]. In addition, with the aim of mimicking a medicine with prolonged activity, a longer antineoplastic effect (20 days) was tested by administering CBD-Nem 20 and 50 µM. The formulation demonstrated inhibition in the colony formation ability of CF41.Mg cells (Figure 7). To the best of our knowledge, there are no other studies using Nem containing CBD, or other CBD formulations comprising only FDA-approved components and demonstrating efficacy in clonogenic studies.

In terms of the cellular phase, it was also observed that CF41.Mg cells treated with CBD (whether formulated in ethanol or Nem) remained in a higher proportion in the G0–G1 phase, with no significant differences compared to the control (Figure 8A). Additionally, a reduction in the proportion of CF41.Mg cells in the G2-M phase were noted, which suggests a cell cycle arrest at the S phase. Concordantly, an increase in the S phase was observed in response to 50 µM CBD-Nem, which indicates an arrest of DNA replication. In this context, the intracellular DNA repair mechanism is activated, triggering phosphorylation of the CDK2/Cyclin E complex, which leads to prolongation of the S phase and would provide more time for DNA repair [58]. Previous studies in human breast cancer cell lines (MCF-7, ZR-75-1, T47D, MDA-MB-231, MDA-MB-468, and SK-BR3) have shown that treatment with CBD delivered in DMSO induces G0-G1 arrest through CB1 receptor activation, leading to downregulation of cyclin D1 [55,59,60]. Additionally, activation of the CB2 receptor has been reported to affect cell cycle progression by inhibiting Cdc2, a key regulator of cell division, thereby inducing cell cycle arrest in the G2/M phase [60,61,62]. Another study demonstrated that CBD arrests the cell cycle of MDA-MB-231 cells in the late S-G2 phase rather than in G1 [63]. Consistent with these findings, CBD has been shown to exert antiproliferative activity by inducing cell cycle arrest at different phases depending on the concentration used, suggesting the involvement of distinct mechanisms of action underlying its antiproliferative effects [60,63]. In the present work, we demonstrate that this biological potential for CBD during cellular phases is maintained, although the drug is formulated in the form of CBD-Nem.

Moreover, a significant increase in apoptosis in CF41.Mg cells treated with 20 and 50 µM CBD-Nem for 24 h were evidenced; this increase is similar to that observed when using the positive control doxorubicin (1 µM) (Figure 8B). Interestingly, these results contrast with those obtained at 24 h in the MTS reduction assay (Figure 4A), which showed a non-significant 5% decrease in CF41.Mg cell viability following treatment with CBD-Nem at 20 µM. Considering that both assays evaluate distinct cellular processes (MTS is associated with cellular respiration, while the apoptosis assay detects extracellular exposure of phosphatidylserine on the cell membrane), the results may reflect biological overlapping processes at the 24 h time point. Importantly, the initiation of apoptosis can be reversed if the cell manages its homeostatic balance [64]. In another study conducted by Henry et al., 2020 [10], the canine mammary carcinoma cell line CMT12 treated with 11 µM of CBD (in ethanol) showed a significant increase in annexin V-positive cells after 8 h. Furthermore, CBD delivered in DMSO has been reported to induce an apoptotic effect involving caspase-3 in MDA-MB-231 cells [55]. Additional evidence indicates that CBD exerts pro-apoptotic effects by downregulating mTOR, AKT, 4EBP1, and cyclin D, while upregulating the expression and nuclear translocation of PPAR [13,59].

As shown in Figure 9, the exposure to CBD-Nem 20 and 50 µM also prevents migration and invasion in CF41.Mg cells. Similar effects have been observed in human cell lines 4T1.2 and SUM159 treated with 6 µM CBD dissolved in DMSO [8]. McAllister et al. [65] described that CBD negatively regulates the production of ROS and the Id-1 protein in MDA-MB-231 cells, producing a decrease in migration and invasion. It was also observed that metalloproteinases, phalloidin, and actin stress fibers, which are important in tumor invasion, were suppressed by CBD [65]. However, the specific CBD receptors involved in these effects are unknown.

Although this study shows that our CBD-Nem has antitumor effects on canine mammary cancer cells, further research is needed to elucidate its therapeutic potential. The uptake/internalization study could be improved by using antibodies against cannabidiol to directly label it, as well as antibodies for co-localization of endosomal pathways, such as anti-EEA1 and anti-Rab5, among others. These studies could be complemented with mechanistic assays to investigate the pathways underlying the effects of CBD-Nem. On the other hand, in vivo pharmacokinetic studies of CBD-Nem in tissues and animal models should be conducted to evaluate the concentrations reached and the persistence of the formulation in tissues. All this information supports the performance of clinical trials in dogs with mammary carcinomas to define the safety and efficacy of this formulation.

## 5. Conclusions

To our knowledge, this is the first study to evaluate the effects of CBD incorporated into nanoemulsion (CBD-Nem) on mammary canine cancer cells, with results that could support future clinical trials. Our findings indicate antineoplastic activity at concentrations comparable to those previously studied in human breast cancer cells. Importantly, in this study, CBD-Nem was obtained without the use of solvents such as ethanol or DMSO, employing low-energy techniques that are easily scalable. These formulations exhibited high stability and sustained release, potentially resulting in prolonged effects that could reduce the need for repeated administration. The above hypothesis is supported within this paper because CBD-Nem prevented the colony formation (20 days) of canine cancer cells. CBD-Nem demonstrated preferential toxicity for cancer (CF41.Mg) over non-cancer (MDCK) cells. Furthermore, CBD-Nem prevents cell migration and invasion, a key pathological aspect of tumor malignancy. Further research is needed to elucidate molecular mechanisms underlying the observed effects and the therapeutic potential in canine mammary cancer.

## Figures and Tables

**Figure 1 pharmaceutics-17-00970-f001:**
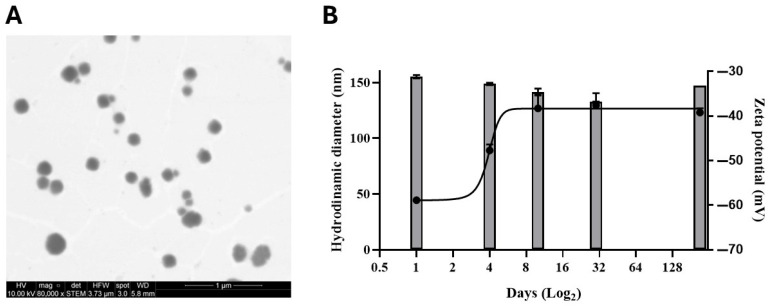
CBD-Nem characterization (morphology, stability, and release). (**A**) Scanning transmission electron microscopy (STEM) image shows homogeneous and spherical nanovehicles of CBD-Nem. (**B**) Figure shows the CBD-Nem stability up to 215 days (hydrodynamic diameter is represented by bars and zeta potential by a continuous line) (n = 3, mean ± S.D).

**Figure 2 pharmaceutics-17-00970-f002:**
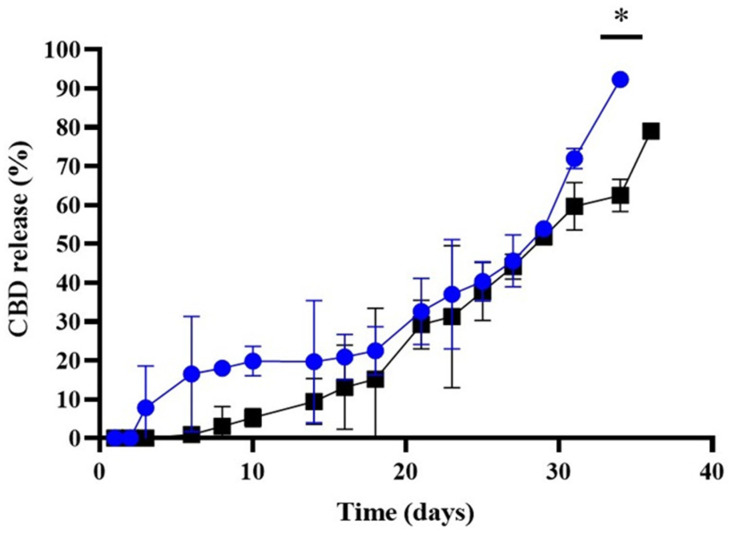
CBD release assay. In vitro cumulative release profile of CBD from CBD-Nem in PBS pH 5.5 (blue lines) and PBS 7.4 (black line) plus polisorbate 80 at 37 °C (n = 3, mean ± S.D., * *p* = 0.0006 at day 34).

**Figure 3 pharmaceutics-17-00970-f003:**
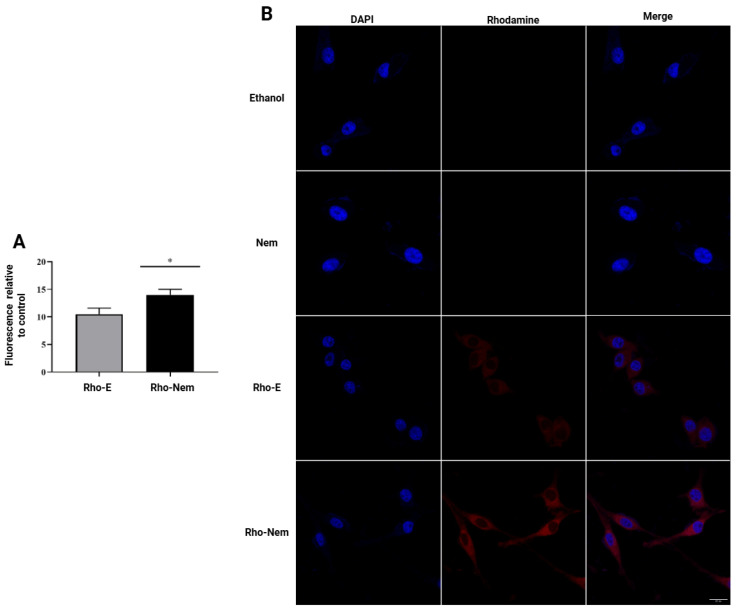
Cell uptake and internalization of Rho-Nem in CF41.Mg cells. (**A**) Quantification of the mean fluorescence intensity relative to controls (MFRC) of Rho-E and Rho-Nem (n = 3, mean ± S.D., * *p* < 0.001). (**B**) Representative microphotographs of CF41.Mg cells by confocal fluorescence microscopy after applying Rho-E and Rho-Nem (red fluorescence) and staining nuclei (DAPI staining blue color) (magnification ×40), n = 3. Scale bar = 20 µm.

**Figure 4 pharmaceutics-17-00970-f004:**
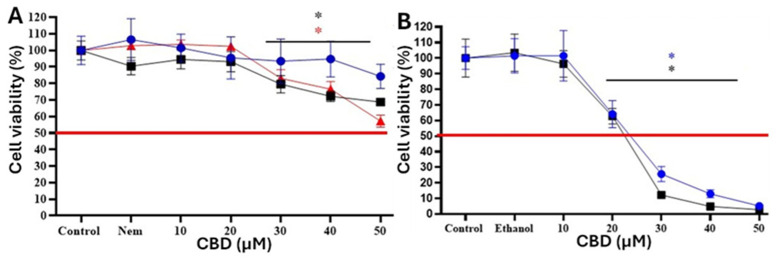
Cell viability (MTS) of mammary carcinoma cells CF41.Mg in response to CBD-Nem and CBD-E. (**A**) CF41.Mg cells treated with CBD-Nem for 24 (blue), 48 (black), and 72 h (red) (n = 3, mean ± S.D., * *p* < 0.0001 at 48 h in relation to control (culture medium), * *p* < 0.0001 at 72 h in relation to control). (**B**) CF41.Mg cells treated with CBD-E for 24 (blue) and 48 h (black) (n = 3, mean ± S.D., * *p* < 0.0001 at 24 h in relation to control; * *p* < 0.0001 at 48 h in relation to control).

**Figure 5 pharmaceutics-17-00970-f005:**
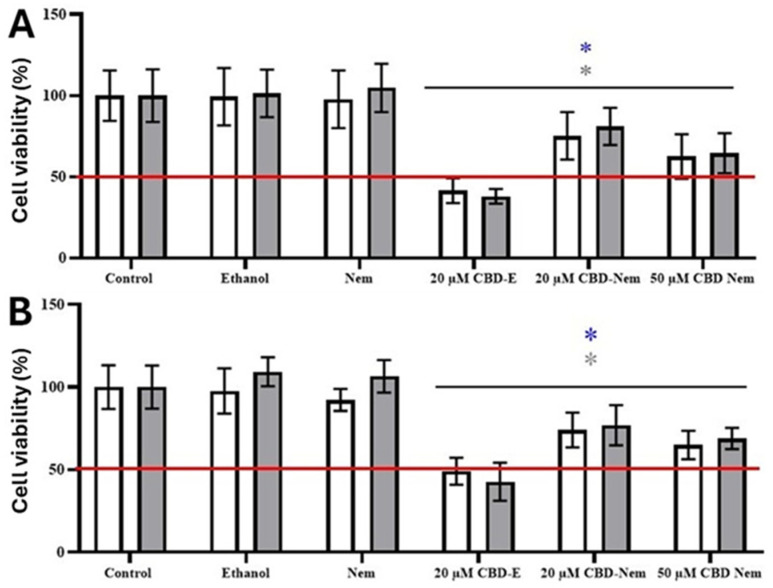
Viability (MTS) of mammary carcinoma cells IPC366 and MDA-MB-231 cells in response to CBD-E 20 µM of, CBD-Nem 20 and 50 µM. (**A**) IPC366 cells treated for 48 (white) and 72 h (grey). (n = 3, mean ± S.D., * *p* < 0.0001 at 48 h in relation to control, * *p* < 0.0001 at 72 h in relation to control (culture medium)). (**B**) MDA-MB-231 cells treated for 48 (white), and 72 grey hours (grey). (n = 3, mean ± S.D., * *p* < 0.0001 at 48 h in relation to control, * *p* < 0.0001 at 72 h in relation to control).

**Figure 6 pharmaceutics-17-00970-f006:**
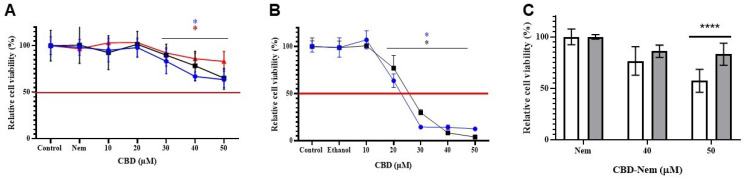
Viability (MTS) of MDCK canine renal epithelial cells exposed to CBD-Nem (**A**) and CBD-E (**B**); comparison of the viability between CF41.Mg and MDCK cells treated with CBD-Nem (**C**). (**A**) MDCK cells were treated with CBD-Nem for 24 (blue), 48 (black), and 72 h (red) (n = 3, mean ± S.D., * *p* < 0.0001 at 24 h in relation to control (culture medium), * *p* < 0.0001 at 48 h in relation to control, * *p* < 0.0001 at 72 h in relation to control). (**B**) MDCK cells were treated with CBD-E for 24 (blue) and 48 h (black) (n = 3, mean ± S.D., * *p* < 0.0001 at 24 h in relation to control (culture medium), * *p* < 0.0001 at 48 h in relation to control). (**C**) Comparison of cell viability relative to Nem between CF41.Mg (white) and MDCK (grey) cells in response to CBD-Nem at concentrations of 40 and 50 µM at 72 h (**** *p* < 0.0001).

**Figure 7 pharmaceutics-17-00970-f007:**
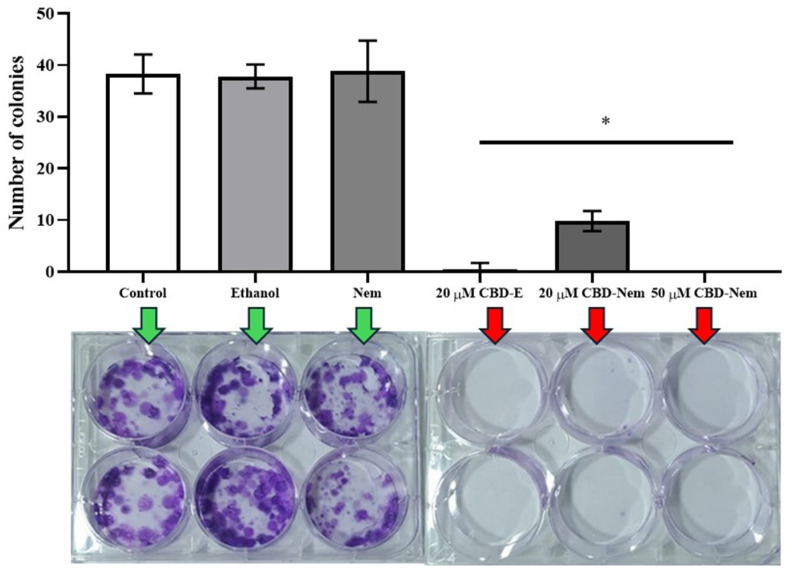
Clonogenic assay in CF41.Mg cells treated with CBD-Nem. Quantification of the number of cell colonies after 20 days of culture. Experimental conditions: control (culture medium), ethanol, Nem, CBD-E 20 µM, and CBD-Nem 20 and 50 µM (n = 3, mean ± S.D., * *p* ≤ 0.0001 relative to the control). Below the graph, a representative photograph of the plaque at day 20 can be appreciated (each column represents a duplicate in the same plaque).

**Figure 8 pharmaceutics-17-00970-f008:**
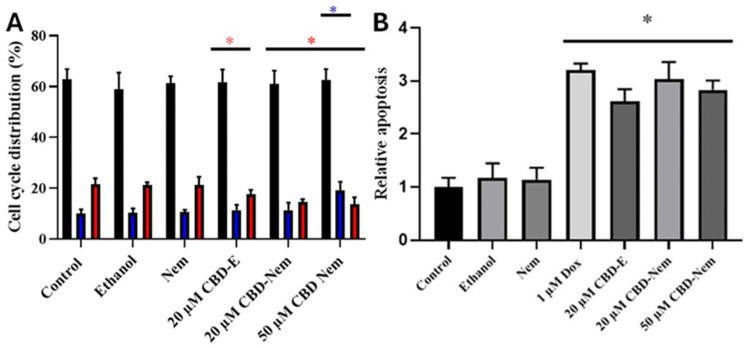
(**A**) Cell cycle assay in CF41.Mg cells treated with CBD-Nem. Experimental conditions: control (culture medium), ethanol, Nem, CBD-E 20 µM, CBD-Nem 20 µM and 50 µM, (black = G0-G1; blue = S; red = G2-M) (n = 3 (duplicate), mean ± S.D., * *p* ≤ 0.0001 compared to the control. * *p* ≤ 0.0001 compared to the control for CBD-Nem 20 µM and 50 µM. * *p* = 0.0174 compared to the control for CBD-E 20 µM). (**B**) Apoptosis assay in CF41.Mg cells treated with CBD-Nem. The graph shows the quantification of apoptosis relative to the control (culture medium) in CF41.Mg cells in response to 1 µM doxorubicin, 20 µM CBD-E, and 20 or 50 µM CBD-Nem (n = 4 (duplicate), mean ± S.D., * *p* ≤ 0.0001 compared to the control).

**Figure 9 pharmaceutics-17-00970-f009:**
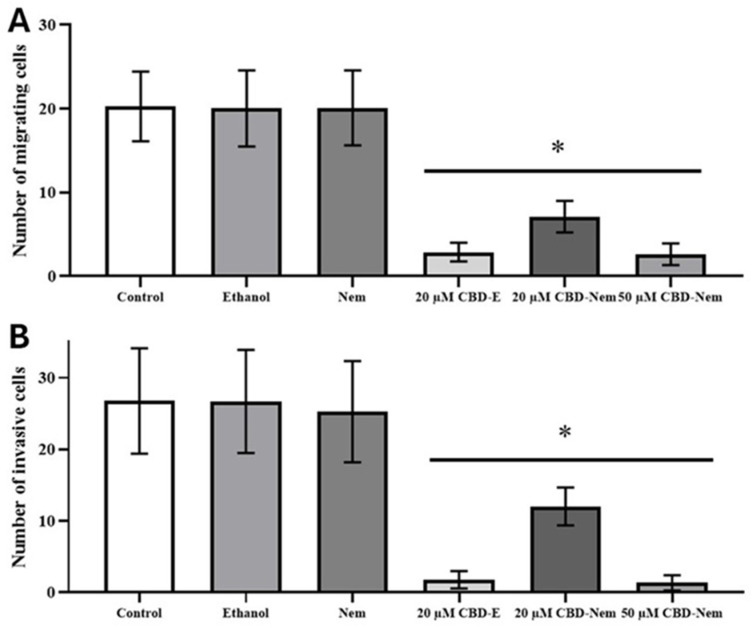
Migration and invasion ability of CF41.Mg cells in response to CBD-Nem. (**A**) Shows the quantification of the number of migrated cells in response to CBD-E 20 µM and CBD-Nem 20 and 50 µM for 6 h (n = 3 (duplicate), mean ± S.D., * *p* < 0.0001 relative to the control (culture medium)). (**B**) Shows the quantification of the number of invasive cells in response to CBD-E 20 µM, CBD-Nem 20 and 50 µM for 24 h (n = 3 (duplicate), mean ± S.D., * *p* < 0.0001 relative to the control (culture medium)).

**Table 1 pharmaceutics-17-00970-t001:** Characterization of nanoemulsions. Particle size, polydispersity, zeta potential, nanoemulsion concentration, and CBD-Nem encapsulation efficiency. A CBD concentration of 0.1% *w*/*v* was added to the formulation. Results are presented as the mean of three measurements (n = 3) with the corresponding standard deviation.

	Hydrodynamic Diameter (nm)	Polidispersity Index (PDI)	Zeta Potential (mV)	Nem Concentration (Nanoparticles/mL)	Encapsulation Efficiency
Nem	150 ± 9	0.14 ± 0.1	−51 ± 2	1.2 × 10^13^ ± 4^13^	
CBD-Nem	155 ± 12	0.14 ± 0.1	−53 ± 7	4.8 × 10^13^ ± 6^13^	≥99%
Rhod-Nem	138.1 ± 7	0.13 ± 0.1	−56 ± 7		

## Data Availability

The data that support the findings of this study are available from the corresponding author upon reasonable request.

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
