# Peer review of "Formulation and Functional Characterization of a Cannabidiol-Loaded Nanoemulsion in Canine Mammary Carcinoma Cells"

_pharmaceutics, 2025, doi:10.3390/pharmaceutics17080970_

Round 1
Reviewer 1 Report
Comments and Suggestions for Authors
The present study aimed to develop CBD-loaded nanoemulsion (CBD-Nem), as well as to determine its antitumor activity on canine (and human) mammary carcinoma cells, its uptake, internalization, apoptosis, colony formation, migration, and invasion in these malignant cells. The study is well-designed, some new and significant results have been obtained, and the manuscript is generally well written. There are, however, some concerns that need to be addressed.
A much lower cytotoxicity of CBD nanoemulsion was demonstrated in comparison to the ethanol solution, and this is one of the main results of the study that has not been sufficiently discussed. IC50 was not reached for CBD-Nem even at 50 µM. Further discussion on this is required.
The authors appear to overstate the therapeutic potential of cannabidiol (CBD) in mammary carcinoma, repeatedly referring to it as a promising or potential treatment option throughout the text. While preclinical data suggest some anticancer activity, there is currently insufficient clinical evidence to support such claims, especially in veterinary oncology. No large, randomized controlled trials have confirmed its effectiveness for treating breast cancer in humans (or mammary carcinoma in dogs). These overstatements (in the abstract, introduction and discussion) need to be rephrased, and the real level of evidence for CBD should be clear in the text.
Furthermore, the text may overemphasize the toxicological risks of ethanol, DMSO or poloxamer 188, which are generally considered safe within commonly used concentration ranges and approved by the FDA as solvents and excipients in oral, topical, and injectable pharmaceuticals.
There are some terminological inconsistencies in the text. For example, the term "synthesis" in the title and in the main text is not suitable. "Synthesis" typically refers to chemical reactions that create new compounds; the terms "preparation", "development" or "formulation" of nanoemulsions are preferred over "synthesis." For example, CBD-Nem was not synthesized, but prepared or developed (line 28).
The main formulation ingredients need to be defined in the abstract (type of the oil and the emulsifier).
The water solubility of CBD should be specified in the Introduction. Also, the solubility of CBD in the MCT oil should be stated; was CBD completely dissolved in small amount of oil in nanoemulsion? Besides, what BCS category does CBD belong to? It should be also discussed in the context of CBD pharmacokinetics.
Pharmacokinetics of CBD needs to be discussed more in-depth in the manuscript. Are these concentrations (20 µM or 50 µM) achievable in vivo (in tissues) following the oral administration of standard doses for example?
How was nanoemulsion concentrated to exactly 5 ml? The final oil concentration in the nanoemulsion should also be specified (was it 2.5%?).
For nanoemulsions, “encapsulation efficiency” is the preferred term over “association efficiency”. I suggest changing it throughout the text.
What is the rationale for studying the drug release for 384 hours? It is usually the case for implants or in situ gels. It should be further discussed in Discussion if some other nanoemulsion formulations exerted such sustained release.
In the Results section, the authors refer to 'solutions' when describing nanoemulsions, which is incorrect and should be revised.
When referring to CBD-Nem, what concentration of CBD is it (for example in Table 1)?
In Discussion, the authors mention the release kinetics for the first time. All models tested need to be specified in the Methods section.
While the approach of using Rho instead of CBD is reasonable for a preliminary, indirect assessment of CBD uptake, the results however should be interpreted cautiously. Rho is a charged molecule, while CBD is neutral, which may affect uptake pathways. Besides, these two molecules have no structural similarity, so mechanisms like carrier-mediated transport or endocytosis may not align. This should be emphasized in the text.
Line 40: it is the first time stability was mentioned (in conclusion).
Line 113: please, specify what Miglyol oil was used.
Comments on the Quality of English LanguageThe manuscript is generally clear, though some minor language issues remain that could be resolved with additional proofreading.
Author Response
We appreciate all the comments, which significantly contribute to improving the manuscript. Below, we detail our responses to each point.
Comment 1: A much lower cytotoxicity of CBD nanoemulsion was demonstrated in comparison to the ethanol solution, and this is one of the main results of the study that has not been sufficiently discussed. IC50 was not reached for CBD-Nem even at 50 µM. Further discussion on this is required.
Response 1: A paragraph has been incorporated to improve the discussion of these less pronounced effects, from lines 594 to 605 in the Discussion section.
Comment 2: The authors appear to overstate the therapeutic potential of cannabidiol (CBD) in mammary carcinoma, repeatedly referring to it as a promising or potential treatment option throughout the text. While preclinical data suggest some anticancer activity, there is currently insufficient clinical evidence to support such claims, especially in veterinary oncology. No large, randomized controlled trials have confirmed its effectiveness for treating breast cancer in humans (or mammary carcinoma in dogs). These overstatements (in the abstract, introduction and discussion) need to be rephrased, and the real level of evidence for CBD should be clear in the text.
Response 2: We agree with these observations. The emphasis on the potential therapeutic use of our formulation has been reduced to better align the manuscript with the current state of research.
Comment 3: Furthermore, the text may overemphasize the toxicological risks of ethanol, DMSO or poloxamer 188, which are generally considered safe within commonly used concentration ranges and approved by the FDA as solvents and excipients in oral, topical, and injectable pharmaceuticals.
Response 3: We agree with the observation. References to the toxicity of solvents such as ethanol and DMSO have been deleted throughout the text.
Comment 4: There are some terminological inconsistencies in the text. For example, the term "synthesis" in the title and in the main text is not suitable. "Synthesis" typically refers to chemical reactions that create new compounds; the terms "preparation", "development" or "formulation" of nanoemulsions are preferred over "synthesis." For example, CBD-Nem was not synthesized, but prepared or developed (line 28).
Response 4: The word "synthesis" has been deleted throughout the text and replaced with "formulation" or "preparation”.
Comment 5: The main formulation ingredients need to be defined in the abstract (type of the oil and the emulsifier).
Response 5: These components have been incorporated into the abstract. Thank you for this comment.
Comment 6: The water solubility of CBD should be specified in the Introduction. Also, the solubility of CBD in the MCT oil should be stated; was CBD completely dissolved in small amount of oil in nanoemulsion? Besides, what BCS category does CBD belong to? It should be also discussed in the context of CBD pharmacokinetics.
Response 6: We have incorporated information on the solubility of CBD, and its biopharmaceutical classification in the Introduction (lines 78–81). Moreover, in the discussion section, its solubility in MCT is described (lines 506-507).
Comment 6: Pharmacokinetics of CBD needs to be discussed more in-depth in the manuscript. Are these concentrations (20 µM or 50 µM) achievable in vivo (in tissues) following the oral administration of standard doses for example?
Response 6: We have added to the manuscript a specification of our proposed route of administration for this nanoemulsion, which would be primarily topical after the surgical removal of the primary tumor, to ensure high concentrations of the CBD-Nem formulation in the tissues (lines 342-347).
Comment 7: How was nanoemulsion concentrated to exactly 5 ml? The final oil concentration in the nanoemulsion should also be specified (was it 2.5%?).
Response 7: The concentration of the ingredients used in the formulation has been included in the Methods section. The sample was concentrated from 20 mL to 5 mL using rotary evaporation, with the volume measured at regular intervals using a graduated cylinder until the desired volume was reached (lines 131-142).
Comment 8: For nanoemulsions, “encapsulation efficiency” is the preferred term over “association efficiency”. I suggest changing it throughout the text.
Response 8: The word "association" has been replaced with "encapsulation" throughout the text.
Comment 9: What is the rationale for studying the drug release for 384 hours? It is usually the case for implants or in situ gels. It should be further discussed in Discussion if some other nanoemulsion formulations exerted such sustained release.
Response 9: The rationale for evaluating this formulation over a 36-day period is its potential use following the surgical removal of the primary tumor, to determine the possible duration of the effect after administration (incorporated into the Results section, lines 342–347). Information on the release profiles of other formulations has been added to the Discussion (lines 542–553).
Comment 10: In the Results section, the authors refer to 'solutions' when describing nanoemulsions, which is incorrect and should be revised.
Response 10: The word "solution" has been removed throughout the text.
Comment 11: When referring to CBD-Nem, what concentration of CBD is it (for example in Table 1)?
Response 11: We appreciate this comment. The concentration of CBD has been incorporated into Table 1 legend.
Comment 12: In Discussion, the authors mention the release kinetics for the first time. All models tested need to be specified in the Methods section.
Response 12: Information on the kinetic modeling used has been incorporated into the Methods section (lines 191-196).
Comment 13: While the approach of using Rho instead of CBD is reasonable for a preliminary, indirect assessment of CBD uptake, the results however should be interpreted cautiously. Rho is a charged molecule, while CBD is neutral, which may affect uptake pathways. Besides, these two molecules have no structural similarity, so mechanisms like carrier-mediated transport or endocytosis may not align. This should be emphasized in the text.
Response 13: We agree with this comment. A paragraph discussing the limitations of using Rho-Nem to indirectly assess uptake/internalization has been incorporated (lines 564–569).
Comment 14: Line 40: it is the first time stability was mentioned (in conclusion).
Response 14: The methodology for stability analysis and its corresponding results have been incorporated into the text (lines 155-158).
Comment 15: Line 113: please, specify what Miglyol oil was used.
Response 15: The type of Miglyol used (812) has been incorporated (line 130).
Reviewer 2 Report
Comments and Suggestions for Authors
Synthesis and functional characterization of a Cannabidiol-loaded nanoemulsion on canine mammary carcinoma cells
In this work a Cannabidiol-loaded nanoemulsion was develop, and evaluate its effects on canine or non-cancer cell lines. The work was carefully done. The conclusions are supported by the results presented. However, major revisions should be made to improve the manuscript
- INTRODUCTION
The authors should include a review of nanoemulsions containing cannabidiol and to detail the innovations of the study.
MATERIAL AND METHODS
- Nanocarriers preparation and characterization
- a) In the cannabidiol dosage analysis, for Association efficiency, and CBD-release, include the detection and quantification limits of the UV methodology.
- b) Include the stability test in the methodology indicating what was evaluated during this test
RESULTS AND DISCUSSION
- Improve the quality of the Figure 1
- Include the results of stability test
Author Response
We appreciate all the comments, which significantly contribute to improving the manuscript. Below, we detail our responses to each point.
INTRODUCTION
Comment 1: The authors should include a review of nanoemulsions containing cannabidiol and to detail the innovations of the study.
R: Thank you for your comment. Additional information on CBD nanoformulation techniques has been incorporated into the Introduction section (lines 89–100).
MATERIAL AND METHODS
Comment 2: Nanocarriers preparation and characterization
a) In the cannabidiol dosage analysis, for Association efficiency, and CBD-release, include the detection and quantification limits of the UV methodology.
Response 2: The limits of detection and quantification for the UV spectrophotometry method have been incorporated into the Methods section (lines 196–198).
b) Include the stability test in the methodology indicating what was evaluated during this test.
Response: The stability study has been included in the Methods section (lines 155–158).
RESULTS AND DISCUSSION
Comment 3: Improve the quality of the Figure 1.
Response 3: Figure 1 has been improved.
Comment 4: Include the results of stability test
Response 4: The results of the stability study have been incorporated (lines 319–323).
Reviewer 3 Report
Comments and Suggestions for Authors
The manuscript developed an oil-in-water nanoemulsions for poor water soluble drug canabidiol, with aim to treat canine mammary carcinoma. It is an interesting topic and may lead to potential application of pet therapy. Some comments are below:
- for the formulation, has selection of excipients been justified? Is there any high-througtput screening data to determine optimal formulation?
- In the drug release figure, CBD-NEM showed a long drug release. In particular, at the early stage (0-7days), the drug release was minimum. How did CBD exert its cytotoxicity if it is not released?
- In Figure 1B, why did the formulation show increased zeta-potential? Is there any aggregation?
- What triggered the drug release? Authors should perform more nanostructure characterization and drug-excipient interaction to determine release mechanism.
- Figure 3, cell uptake image is poor.
- It appears CBD-NEM did not perform better than CBD in ethanol. But the cell uptake study showed the opposite result. why?
- In clonogenic assay, how was colonies quantified? In Figure 7, cell colonies was a spear. Please provide detailed quantification method.
Author Response
We appreciate all comments, which significantly contribute to improving the manuscript. Below, we detail our responses to each point.
Comment 1: For the formulation, has selection of excipients been justified? Is there any high-througtput screening data to determine optimal formulation?
Response 1: The nanoemulsion has been previously developed by our team with other lipophilic compounds such as curcumin. This clarification has been incorporated into the Methods section (line 129) and the Discussion (lines 502–503).
Comment 2: In the drug release figure, CBD-NEM showed a long drug release. In particular, at the early stage (0-7days), the drug release was minimum. How did CBD exert its cytotoxicity if it is not released?
Response 2: The discussion regarding the cytotoxic effect of nanoemulsion despite its slow release has been improved (lines 540-545). The effect observed despite the low release in the first few days could be partially explained by the probable internalization of CBD-Nem, which would release CBD intracellularly and allow its interaction with receptors located within the cell (589-606).
Comment 3: In Figure 1B, why did the formulation show increased zeta-potential? Is there any aggregation?
Response 3: The increase in zeta potential may be due to the degradation of the surfactant in the formulation. However, the literature describes that formulations maintaining zeta potentials below minus thirty millivolts exhibit low aggregation over time, as shown in the stability results now included (lines 319–323).
Comment 4: What triggered the drug release? Authors should perform more nanostructure characterization and drug-excipient interaction to determine release mechanism.
Response 4: We appreciate this comment, and we agree that it would be very interesting to conduct additional studies to define the CBD release mechanism. However, we cannot implement what the reviewer suggested in the short term. CBD shows high compatibility with the oily core of the nanoparticle, which explains the high association of the drug with the excipient. Regarding the most likely mechanism of CBD release, we think that the main mechanism of release would be the degradation of the surfactant compound of the obtained nanoparticles, since we estimated the best kinetic model using the DDSolver add-in for Microsoft Excel., and the release model with the highest fit was the Hopfenberg kinetics model, indicating that the main limiting process in the release of CBD from nanoemulsions was matrix erosion.
Comment 5: Figure 3, cell uptake image is poor.
Response 5: The Quality of Figure 3 has been improved.
Comment 6: It appears CBD-NEM did not perform better than CBD in ethanol. But the cell uptake study showed the opposite result. why?
Response 6: A paragraph has been incorporated into the Discussion addressing the possible causes of the increased internalization of CBD and its low cytotoxic effect (lines 594–598).
Comment 7: In clonogenic assay, how was colonies quantified? In Figure 7, cell colonies was a spear. Please provide detailed quantification method.
Response 7: Additional information required on colony quantification has been incorporated into the Methods section (lines 260–263).
Reviewer 4 Report
Comments and Suggestions for Authors
This manuscript describes the development and in vitro evaluation of a cannabidiol (CBD)-loaded oil-in-water nanoemulsion (CBD-Nem) for its anti-cancer potential in canine mammary carcinoma cell lines. The authors present physicochemical characterization, cell viability, internalization, cell cycle, apoptosis, and migration/invasion assays. The study is well-written and data-rich, but it has some critical limitations that need to be addressed before it can be considered for publication.
- The introduction section is comprehensive, but there is overreliance only on the cannabinoid, there is insufficient discussion of recent nanoemulsion strategies in veterinary oncology. The authors are suggested to expand on translational relevance to veterinary therapeutics.
- The drug release 36-day study seems excessive without justification for relevance to in vitro assays. The authors need to discuss why such prolonged release is evaluated, and whether this aligns with in vivo expectations.
- The discussion section Overstates clinical relevance without adequate preclinical data. The authors should also discuss about the limitation of current research and try to suggest any future studies that could be performed in order to mitigate said limitations.
Author Response
We appreciate all comments, which significantly contribute to improving the manuscript. Below, we detail our responses to each point.
Comment 1: The introduction section is comprehensive, but there is overreliance only on the cannabinoid, there is insufficient discussion of recent nanoemulsion strategies in veterinary oncology. The authors are suggested to expand on translational relevance to veterinary therapeutics.
Response 1: Additional information on the use of nanoemulsions in veterinary medicine has been incorporated into the Introduction (lines 106–107)
Comment 2: The drug release 36-day study seems excessive without justification for relevance to in vitro assays. The authors need to discuss why such prolonged release is evaluated, and whether this aligns with in vivo expectations.
Response 2: The rationale for evaluating this formulation over a 36-day period is its potential use following the surgical removal of the primary tumor, in order to determine the possible duration of the effect after administration (incorporated into the Results section, lines 342–347). Information on the release profiles of other formulations has been added to the Discussion (lines 542–545).
Comment 3: The discussion section Overstates clinical relevance without adequate preclinical data. The authors should also discuss about the limitation of current research and try to suggest any future studies that could be performed in order to mitigate said limitations.
Response 3: The Potential clinical relevance of our study has been modulated, emphasizing the need for further clinical studies to define the clinical potential of the nanoformulation developed here. Limitations of the techniques used and possible improvements for future studies have been incorporated into the Discussion (lines 532–535 and 564–569, 667-676).
Reviewer 5 Report
Comments and Suggestions for Authors
Dear authors: this study provides a well-structured investigation of CBD-loaded nanoemulsions for canine mammary carcinoma treatment. The methods are clearly described, and the results support the conclusions. The presentation is clear, though some figures could benefit from improved labeling. The research holds significance for veterinary medicine and may have broader implications. Minor refinements to the discussion of mechanisms and potential clinical translation would strengthen the paper further. Overall, it represents a solid contribution to the field.

Author Response
Comment 1: Minor refinements to the discussion of mechanisms and potential clinical translation would strengthen the paper further. Overall, it represents a solid contribution to the field.
Response 1: We appreciate this comment. In the discusión section, we had added a paragraph describing the potential clinical translation of our findings (lines 668-677).
Round 2
Reviewer 1 Report
Comments and Suggestions for Authors
My previous suggestions have been mostly addressed, and the manuscript has been improved accordingly. I support its acceptance and recommend it for publication.
Comments on the Quality of English LanguageThere are still some minor language issues that could be resolved with additional proofreading.
Reviewer 2 Report
Comments and Suggestions for Authors
Thank you for the review, I believe the paper can now be accepted for publication.